# Outstanding Radiation Tolerance of Supported Graphene: Towards 2D Sensors for the Space Millimeter Radioastronomy

**DOI:** 10.3390/nano11010170

**Published:** 2021-01-11

**Authors:** Alesia Paddubskaya, Konstantin Batrakov, Arkadiy Khrushchinsky, Semen Kuten, Artyom Plyushch, Andrey Stepanov, Gennady Remnev, Valery Shvetsov, Marian Baah, Yuri Svirko, Polina Kuzhir

**Affiliations:** 1Institute for Nuclear Problems of Belarusian State University, Bobruiskaya Str. 11, 220006 Minsk, Belarus; kgbatrakov@gmail.com (K.B.); arluchr@mail.ru (A.K.); semen_kuten@list.ru (S.K.); artyom.plyushch@gmail.com (A.P.); polina.kuzhir@gmail.com (P.K.); 2Radiophysics Department, Tomsk State University, Lenin Ave, 36, 634050 Tomsk, Russia; 3Faculty of Physics, Vilnius University, Sauletekio 9, LT-10222 Vilnius, Lithuania; 4Research and Production Laboratory “Pulse-Beam, Electric Discharge and Plasma Technologies”, Tomsk Polytechnic University, Lenin Ave, 30, 634050 Tomsk, Russia; stepanovav@mail.ru (A.S.); remnev06@mail.ru (G.R.); 5Joint Institute for Nuclear Research, Joliot-Curie 6, 141980 Dubna, Russia; shv@nf.jinr.ru; 6Institute of Photonics, University of Eastern Finland, P.O. Box 111, FI-80101 Joensuu, Finland; marian.baah@uef.fi (M.B.); yuri.svirko@uef.fi (Y.S.)

**Keywords:** graphene, terahertz, absorption, ionizing radiation, geostationary orbit

## Abstract

We experimentally and theoretically investigated the effects of ionizing radiation on a stack of graphene sheets separated by polymethyl methacrylate (PMMA) slabs. The exceptional absorption ability of such a heterostructure in the THz range makes it promising for use in a graphene-based THz bolometer to be deployed in space. A hydrogen/carbon ion beam was used to simulate the action of protons and secondary ions on the device. We showed that the graphene sheets remain intact after irradiation with an intense 290 keV ion beam at the density of 1.5 × 10${}^{12}$ cm${}^{-2}$. However, the THz absorption ability of the graphene/PMMA multilayer can be substantially suppressed due to heating damage of the topmost PMMA slabs produced by carbon ions. By contrast, protons do not have this negative effect due to their much longer mean free pass in PMMA. Since the particles’ flux at the geostationary orbit is significantly lower than that used in our experiments, we conclude that it cannot cause tangible damage of the graphene/PMMA based THz absorber. Our numerical simulations reveal that, at the geostationary orbit, the damaging of the graphene/PMMA multilayer due to the ions bombardment is sufficiently lower to affect the performance of the graphene/PMMA multilayer, the main working element of the THz bolometer, which remains unchanged for more than ten years.

## 1. Introduction

THz fingerprints of space objects carry information of tremendous importance enabling insight into the past and future of the Universe (see [1] and refs therein). Detecting and analyzing them shed light on the formation and evolution of galaxies and allows one to address fundamental problems of physics of pulsars and quasars [2,3], the formation of black holes and accretion of the surrounding black matter [4,5], microwave amplification by stimulated emission in interstellar and circumstellar molecules [6], and the gravitational waves [7].

Detection of the ultra-weak THz signals currently relies on bolometers [8,9,10,11,12] that convert the electromagnetic energy into heat due to the Joule effect. THz bolometers for space telescopes require materials combining high THz absorption ability, tunability of the detection wavelength in a wide range and high resistance of the bolometer to ionizing radiation.

The strong absorption ability of graphene in the microwave and THz range [13,14,15] makes this material very attractive for use in the THz bolometers. Moreover, it has been shown both numerically [16,17] and experimentall [18,19,20] that graphene is highly resistant to ionizing radiation because the one-atom thick graphene sheet is hardly visible for high energy ionizing particles. Specifically, the bombardment by ions having energy from 0.1 to 90 keV resulted in the formation of 20–60 A${}^{2}$ wide defect areas in the graphene sheet [21], which are too small to influence the THz response. Both rigorous coupling wave analysis [22,23] and experiment [24] have confirmed that if the lateral dimension of each vacancy defect is much smaller than the wavelength (i.e., 300 microns at 1 THz), and defects occupy up to 10% of graphene area, the THz absorptance of the graphene sheet remains the same.

Strong absorption ability and high resistance to ionizing radiation make graphene a material of choice for THz detecting/sensing in space. However, the high tolerance of graphene to ionizing radiation does not necessarily mean that graphene-based devices are also tolerant. This is because implementing graphene into an optoelectronic device implies placing it on dielectric support or by stacking several graphene/dielectric bi-layers to scale the absorption ability [14,15]. Being virtually transparent for the THz radiation these dielectric spacers in such a sandwich structure may be strongly affected by the ionizing radiation.

In this paper, we focus on the robustness against the ionizing radiation of the graphene/ polymer multilayer that can be employed as a “thermal reservoir” to be heated up due to THz absorption. This thermal reservoir complemented with an element capable to change its electrical resistivity with temperature can be seen as a generic THz bolometer device [25]. From the point of view of the bolometer performance, the radiation tolerance implies not only the ability of graphene/polymer multilayer to sustain the ionizing radiation, but also to preserve its THz absorptance under ions’ bombardment. We perform the investigation of the radiation tolerance of graphene/polymethyl methacrylate (PMMA) sandwich which possess outstanding THz absorption ability [14,15] and can be fabricated by using conventional graphene handling technique [26]. By using hydrogen/carbon ion beam we study experimentally the performance of the graphene/PMMA multi-stacks comprising one, three and five graphene/PMMA bi-layers placed onto the fused silica substrate. The numerical modeling of the long-term irradiation of these sandwich structures with proton beam is performed using the Stopping and Range of Ions in Matter package (SRIM-2013, http://www.srim.org) [21,27].

We aim at the evaluation of the graphene/PMMA sandwich sustainability at the geostationary orbit (GEO), which is the focus orbit for THz space telescopes. At the GEO, the protons at the flux density is (1÷10)×10${}^{7}$ cm${}^{-2}$ s${}^{-1}$ MeV${}^{-1}$ dominate the high energy ions. However, heavier ions, which may occur under proton bombardment of the neighboring equipment/structures of the satellite, are also should be considered.

## 2. Materials and Methods

Fabrication of the graphene/PMMA sandwiches have been described elsewhere [13]. Briefly, the CVD polycrystalline graphene was synthesized at 1000 °C in methane atmosphere on catalytic Cu foil. The graphene was spin-coated by 1 μm thick PMMA layer. After that, Cu foil was removed by the wet etching in ferric chloride. After drying the obtained graphene/PMMA bi-layer is placed on the 0.53 mm thick silica substrate. The next bi-layer was fabricated in the same manner and is placed at the top of the first one. Repeating this process allows us to arrive to a 2×2 cm${}^{2}$ sandwich containing several graphene sheets separated by polymer slabs (see Figure 1a for schematic representation).

Using graphene/PMMA sandwich structures that combine highly absorptive graphene sheets and PMMA slabs, simplifies the bolometer fabrication process because the PMMA is conventionally utilized for CVD graphene transfer and handling. Since PMMA is widely employed in nano- and microfabrication, the effect of ion bombardment on this polymer has been extensively studied [28,29]. In particular, it was shown that the low-dose ion implantation causes mostly chain scissions, which leads to the lowering of the PMMA molecular weight. An increase of the implantation dose activates cross-linking processes and may convert PMMA into graphitized material. The reported conductivity of high-dose implanted PMMA is around 10${}^{-4}$ S/cm [29]. However, even in this worst case, the conductivity is too small to affect significantly the high-frequency (THz) electromagnetic response of several μm film thickness. For the bolometer application PMMA works only as a separation layer between two graphene layers and does not influence the system’s total electromagnetic response.

In order to reveal how the ion bombardment changes the THz performance of the PMMA/graphene sandwich we employ the home-made TEMP-4 ions [30] capable to produce ion beam comprising carbon and hydrogen (70% and 30%, respectively) ions guided by a family of magnetically isolated diodes. The ion pulse duration at half-maximum was about 80 ns at the repetition rate of 0.125 Hz, the beam density was $1.5\times 10^{12}$ ions per cm${}^{2}$ at the average ion energy of 290 keV. The focusing system provides a current density up to 10 A·cm${}^{-2}$ at the beam cross-section in the focal plane of 4×4 cm${}^{2}$. The beam intensity can be varied by changing the position of the sample with respect to the focus of the beam.

In order to visualize the effect of the ion bombardment on the macroscopic continuity, homogeneity, and defectiveness of the PMMA/graphene sandwich, we employed Nanofinder High End (Tokyo Instruments, Tokyo, Japan) confocal Raman system with 600 lines per mm grating and 532 nm laser excitation.

The electromagnetic response of graphene/PMMA sandwiches was investigated using transmission mode of time-domain THz spectrometer described in [14].

## 3. Results and Discussion

To study the dependence of graphene/PMMA sandwich properties on the radiation dose we initially irradiated the whole sample with two ion pulses. After that, we protected half of the sample from the ionizing radiation with a copper foil and irradiated the unprotected sample area again by two ion pulses (see Figure 1a).

One can observe from Figure 2 that Raman spectra of the sandwich composed of 5 graphene/PMMA bi-layers show the pronounced G mode peak in the vicinity of 1590 cm${}^{-1}$ and 2D mode peak at 2690 cm${}^{-1}$ before and after irradiation with an ion beam. A very weak D mode at 1350 cm${}^{-1}$, which is associated with defects formed during graphene growth, transfer and fabrication of multilayer structure, confirms the high quality of the CVD graphene [31]. The spectral features in the vicinity of 2800–3000 cm${}^{-1}$ and a weak peak at 1450 cm${}^{-1}$ are associated with C-H stretching and bending vibrations, respectively, in pure PMMA.

In practice, the ratio of $I_{D}$/$I_{G}$ is often used for quantifying the defect density in graphene. Following the relation proposed in [32] the defect density $n_{D}$ can be approximated as
(1)$$n_{D}\left({\mathrm{cm}}^{-2}\right)=10^{14}/\left(\pi L_{D}^{2}\right),$$
where $L_{D}^{2}($nm${}^{2})=\left(1.8\;\pm \;0.5\right)\;\times \;10^{-9}\lambda_{L}^{4}{\left(I_{D}/I_{G}\right)}^{-1}$ is the square of the average distance between defects, $\lambda_{L}$ (in nm) is excitation laser wavelength. As can be seen for graphene before ion modification, the ratio $I_{D}$/$I_{G}$ is approximately 0.45, which corresponds to the defect density ∼10${}^{11}$ cm${}^{-2}$ with the average distance between defects $L_{D}>18$ nm.

We also observed a significant decrease in Raman signal intensity when the irradiation dose increases. Moreover, the reduction in the relative peak intensity of the PMMA bands after irradiation with four ion pulses indicates breaking of chemical bonds within PMMA and decreasing of hydrogen contents in the polymer layer [29]. On the other hand, insignificant change in the relative intensity of D mode and the absence of an additional peak in the vicinity of 1610 cm${}^{-1}$ (so-called D’ mode) after irradiation with two and four ion pulses indicates a relatively weak degradation of the graphene as it is. This can be caused by the destruction and vanishing of the topmost graphene/polymer bi-layer and preserving the same quality of the remaining graphene sheets. Further, this observation will be supported by THz probing of graphene/PMMA sandwiches before and after ions exposure.

Figure 3a–c show the transmittance spectra of graphene/PMMA sandwiches before and after irradiation with the ion beam. The oscillations of the transmittance are caused by interference in the 0.53 mm-thick silica substrate (dielectric constant of silica is 3.5). In the frequency range of 0.2–1.4 THz, silica reflects about 20% of the incident radiation at negligible absorption. Since the PMMA spacers are transparent for the THz radiation, the THz losses in the graphene/PMMA sandwich placed on the fused silica substrate are due to graphene sheets. One can observe from Figure 3d that untreated samples comprising 1, 3 and 5 graphene/PMMA bi-layers absorb 35%, 45% and 55% of radiation at a frequency of 0.6 THz, respectively.

One may expect that since the THz response of the graphene/PMMA sandwich is governed by graphene sheets and silica substrate rather than polymer slabs, the structure should withstand radiation influence as graphene is radiative resistant. However, our experiments reveal that THz transmittance increases by 10–20% after four pulses. Specifically, after irradiation with four ion pulses the THz transmittance of five graphene/PMMA bi-layers is close to that of virgin three graphene/PMMA bi-layers, while transmittance of three irradiated graphene/PMMA bi-layers becomes comparable with that of a single graphene/PMMA bi-layer (see Figure 3d). In accordance with Raman characterization this experimental finding might indicate that irradiation with four ion pulses is sufficient to evaporate or to fly off two topmost graphene sheets, whereas the third and deeper graphene sheets remain untouched.

In order to understand the influence of the ions beam on the graphene/PMMA sandwich structure we calculate the range of both protons and carbon ions in PMMA using the SRIM-2013 package (http://www.srim.org/) as a function of the ion energy. One can observe from Figure 4 that the proton range in PMMA is proportional to the square of the proton energy (parabolic black solid line). The dependence of the carbon ion range on its energy is more complicated.

This can be explained by the fact that dependence of proton energy *E* on the propagation coordinate *x* can be described by the Bohr equation [33]:(2)$$\frac{\partial E}{\partial x}=-\frac{2\pi e^{4}z^{2}n_{e}}{E}\frac{M}{m_{e}}\ln\frac{4Em_{e}}{\langle I\rangle M},$$
where *z* and *M* are the charge and mass of the ion, $m_{e}$ is the electron mass, $n_{e}=ZN_{A}\rho /A$ is the electron density in the medium. Here, $N_{A}$, *Z*, *A* and ρ are the Avogadro constant, atomic number, atomic weight, and density of the medium, respectively, 〈I〉≈13.6Z eV is the ionization potential of the medium.

The solution of Equation (Equation 2) gives the ions mean free path as
(3)$$L=-\frac{E_{0}^{2}m_{e}}{4\pi e^{4}z^{2}n_{e}M\ln\frac{4E_{0}m_{e}}{\langle I\rangle M}}$$
where $E_{0}$ is the initial ion energy. Since the Bohr Equation (Equation 2) implies $4Em_{e}/\langle I\rangle M>1$, at $E_{0}=290$ KeV it is valid for protons and cannot be used to describe energy loss by carbon ions.

Figure 5 demonstrates beam propagation through 5 bi-layers. Table 1 collects results of numerical simulation by SRIM-2013 characterizing ion beam interaction with graphene/PMMA structures comprising one, three and five graphene/PMMA bi-layers. One can observe that penetration depths of carbon and hydrogen ions with an energy of 290 keV into PMMA are 1.25 μm and 4.5 μm, respectively. Therefore, carbon ions propagate till the second PMMA layer and hydrogen ions (protons) are capable to achieve SiO${}_{2}$ substrate when the sandwich structure consists of three graphene/PMMA bi-layers.

Figure 6 shows spatial distribution of implanted ions and vacancies, energy losses due to generation of phonons and ionization in the sandwiches comprising of three (left column) and five (right column) graphene layers.

As anticipated (see Figure 5 and Table 1), the carbon ions reach only the first graphene layer, while the inner graphene sheets are mainly affected by protons (see Figure 6a,b). Since carbon ions (70%) carry the major part of the beam energy, one may expect—and it is supported by the THz measurements—that one graphene layer will be damaged.

By comparing Figure 6c–f, one may see that energy losses due to ionization are 20 times higher than energy losses due to phonon generation. Both processes lead mainly to the heating of the sandwich structure after relaxation.

One can observe from Figure 6e,f that ionization losses of protons in the PMMA have typical Bragg dependence [34], i.e., the losses increase according to Equation (Equation 2) until the moment when the slowed-down proton captures a target electron, after which the ionization losses drop sharply.

Vacancies distribution in the samples are demonstrated in Figure 6g,h. One can see that carbon ions dominate the vacancies formation, while the influence of proton is negligible.

Figure 7 shows the vacancies distribution in 1 μm thick PMMA produced by proton flux in the graphene/PMMA bi-layer predicted by the SRIM-2013 package. According to our calculations the ∼2$\;\times \;10^{-3}$ vacancies per one incident proton are generated in the graphene layer disposed at the back-side of PMMA under proton beam irradiation. Since the proton flux density at GEO is about $10^{8}$ cm${}^{-2}$s${}^{-1}$, one may estimate that about 2% vacancies will be accumulated for ten years. Such a small number of vacancies will not affect the ability of graphene to absorb THz radiation.

It is worth noting that the electron density at GEO is about an order of magnitude higher than that of protons. However, since the proton is three orders of magnitude heavier than the electron, one may expect from Equation (Equation 2) that the electron-induced ionization losses are much lower than those by protons. Moreover, the protons and secondary ions are more efficient in the vacancy production than electrons because the heavier the ion, the more energy it transfers to the nucleus.

Although a low-density proton beam at GEO unlikely increases the medium temperature due to its very low density, the intense beam, especially its carbon part, in our experiment may result in the heating of the irradiated structure. The increase △T of the PMMA spacer temperature under ion irradiation can be estimated from the following equation:(4)$$QE=\rho c_{\upsilon }△TL,$$
where *Q* is the number of ions per unit surface, *E* is the ion energy, ρ, $c_{\upsilon }$ and *L* are the PMMA density, specific heat capacity and the ion mean-free pass. By using the PMMA parameters ρ≈1.2 g/cm${}^{3}$, $c_{\upsilon }\approx 1000$ J/(Kg K), the Equation (Equation 4) gives the temperature of the topmost PMMA spacer $T_{0}∼1000$ K if the initial sample temperature is 300 K. This estimation shows that thermal effects may lead to local melting of the PMMA and/or formation of the gas bubbles within the PMMA slab (compare Figure 1b–d for SEM of graphene/PMMA sandwich before and after irradiation with 2 and 4 pulses, respectively, as well as the sample roughness presented in Figure 1e). Due to a weak adhesion within the graphene/PMMA/SiO${}_{2}$ stack because the Van-der-Waals interaction energy between graphene and PMMA/SiO${}_{2}$ per unit area is as low as 0.4 meV/A${}^{2}$ [35] this local heating leads to the partial destruction of the graphene sandwich samples, as it has been proved by the THz experiment (see Figure 3d).

## 4. Conclusions

The dependence of the THz transmittance and Raman spectra on the radiation dose demonstrates that the irradiation with a beam comprising 30% of hydrogen and 70% of carbon ions having the energy of 290 keV at a density of $1.5\times 10^{12}$ cm${}^{-2}$ due to extensive heating that results in the PMMA spacer melting and formation of the bubbles between graphene/PMMA structures leads to the evaporation of the topmost graphene/PMMA bi-layer. Since the hydrogen ion range in the PMMA is longer than that carbon ion, they could penetrate the inner layers. Our experimental results demonstrate that protons do not cause any noticeable effect on the inner graphene sheets and PMMA spacers. This finding allows us to expect that the proton irradiation will not lead to the degradation of the graphene-based THz devices at the GEO orbit because the proton density there is much lower than that we studied in the experiment. The graphene-based devices will not be also damaged by the electron irradiation, which produces much less ionization in comparison with the proton one (proton mass is 2000 times larger than electron one). It is worth noting that the graphene can be totally protected from the proton irradiation by covering it with the conventional dielectric layer which thickness is much larger (10 microns), or vice versa smaller (100 nm) than proton mean free path in supporting graphene dielectric slab.

Our numerical simulations reveal that at the geostationary orbit, the ions bombardment is insufficient to produce observable damage to the graphene/PMMA multilayer, which should ensure a stable operation of the graphene-based THz bolometer [36,37] for more than ten years. This paves the way for using CVD graphene/PMMA bi-layers to fabricate a new generation of THz bolometers with outstanding ionizing radiation tolerance that could compete with conventionally used CVD diamond-based bolometers [38].

## Figures and Tables

**Figure 1 nanomaterials-11-00170-f001:**
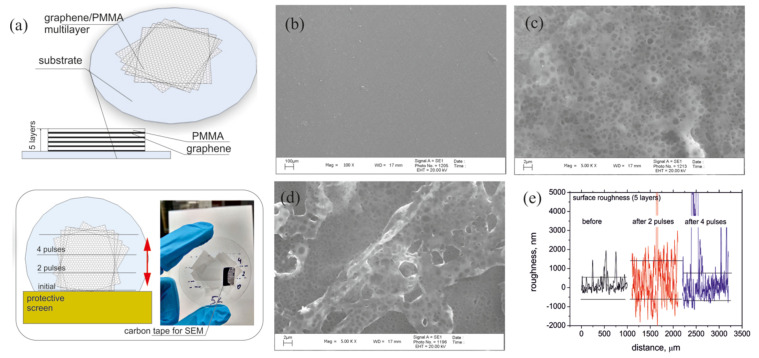
(**a**) Schematic representation of graphene/polymer sandwich on the top of SiO${}_{2}$ substrate and sketch of the ion bombardment experiment, in which increasing of the radiation dose was changed by moving the copper protective screen. Photographic image shows the graphene/polymethyl methacrylate (PMMA) sandwich on the fused silica substrate. (**b**–**d**) SEM images of the sample comprising 5 graphene/PMMA bi-layers before (**b**) and after irradiation with two (**c**) and four (**d**) ion pulses. (**e**) Roughness profile of the samples surface before (black) and after irradiation with 2 (red) and 4 (blue) pulses.

**Figure 2 nanomaterials-11-00170-f002:**
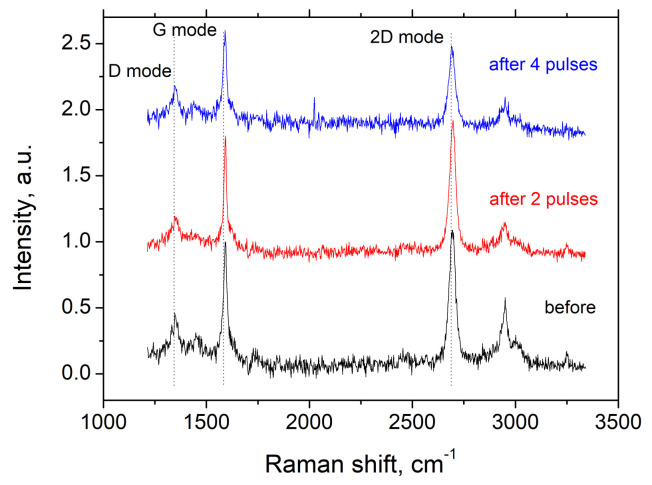
Raman spectra of graphene/PMMA sandwich composed of 5 bi-layers before (black) and after irradiation with two (red) and four (blue) ion pulses having duration of 80 ns and consisting of 70% of carbon and 30% hydrogen ions at the energy of 290 keV. Each spectrum was normalized on the G mode magnitude.

**Figure 3 nanomaterials-11-00170-f003:**
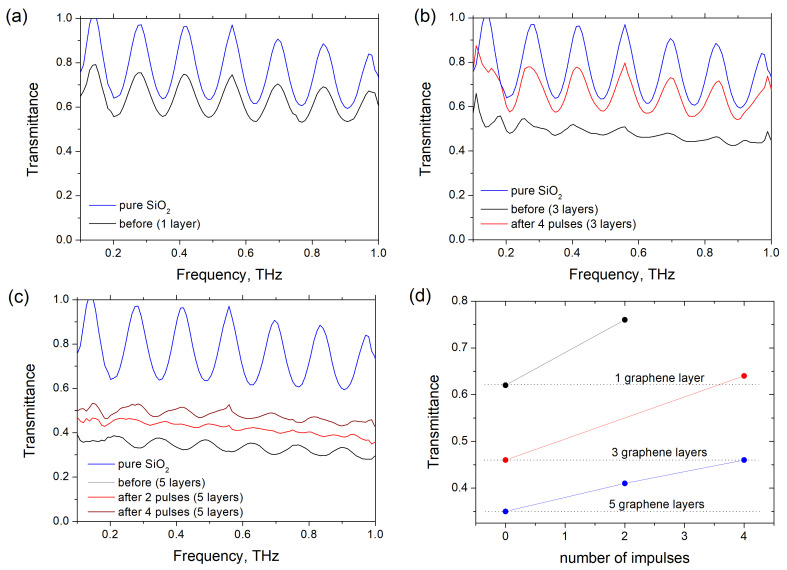
Spectra of the THz transmittance of the graphene/PMMA sandwich comprising 1 (**a**), 3 (**b**) and 5 (**c**) graphene/PMMA bi-layers. Solid lines show transmittance spectra of the silica substrate (blue), virgin sandwich (black), sandwich after irradiation with 2 (red) and 4 (magenta) ion pulses. (**d**) Transmittance of the sample containing one (black dots), three (red dots) and five (blue dots) graphene/PMMA bi-layers at frequency of 0.6 THz vs. the number of ion pulses. Solid lines are the guidance for the eye.

**Figure 4 nanomaterials-11-00170-f004:**
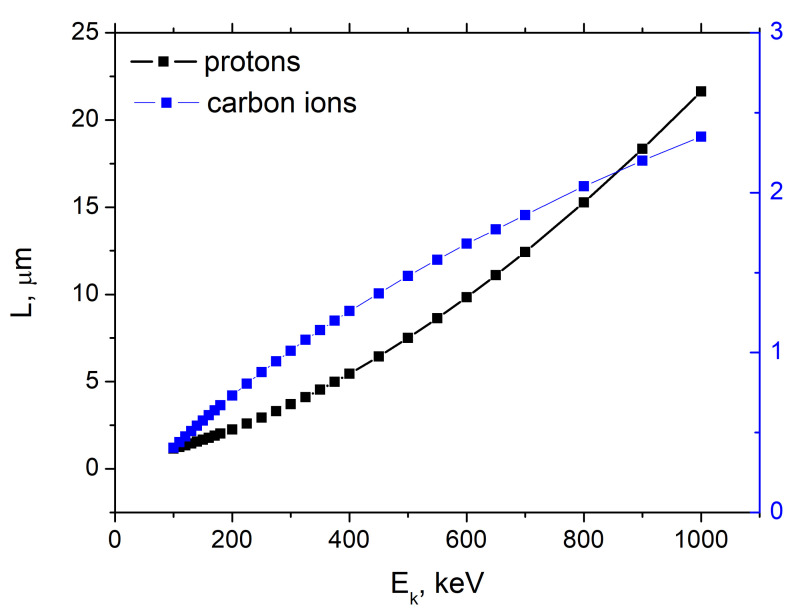
Dependence of the proton (black squares) and carbon ion (blue squares) range in PMMA calculated by SRIM package. The black solid line correspond to Equation (Equation 3).

**Figure 5 nanomaterials-11-00170-f005:**
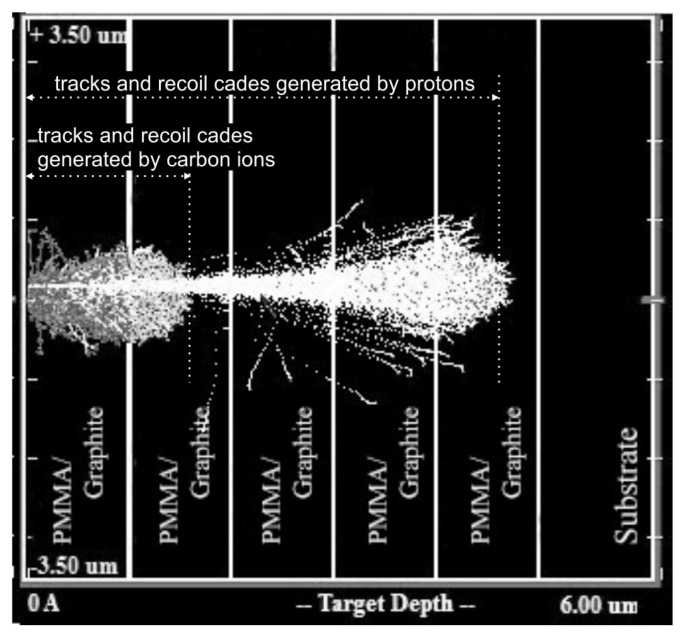
Trajectories of the high-energy particles produced in the structures containing five graphene/PMMA bi-layers under irradiation with the beam comprising 70% carbon ions and 30% protons having the average energy of 290 keV. The thickness of each PMMA spacer is 1 μm. The sandwich structures are deposited on the quartz substrate. One can observe that carbon ions are capable of penetrating into the second PMMA layers, while protons can propagate up to the fourth PMMA spacer.

**Figure 6 nanomaterials-11-00170-f006:**
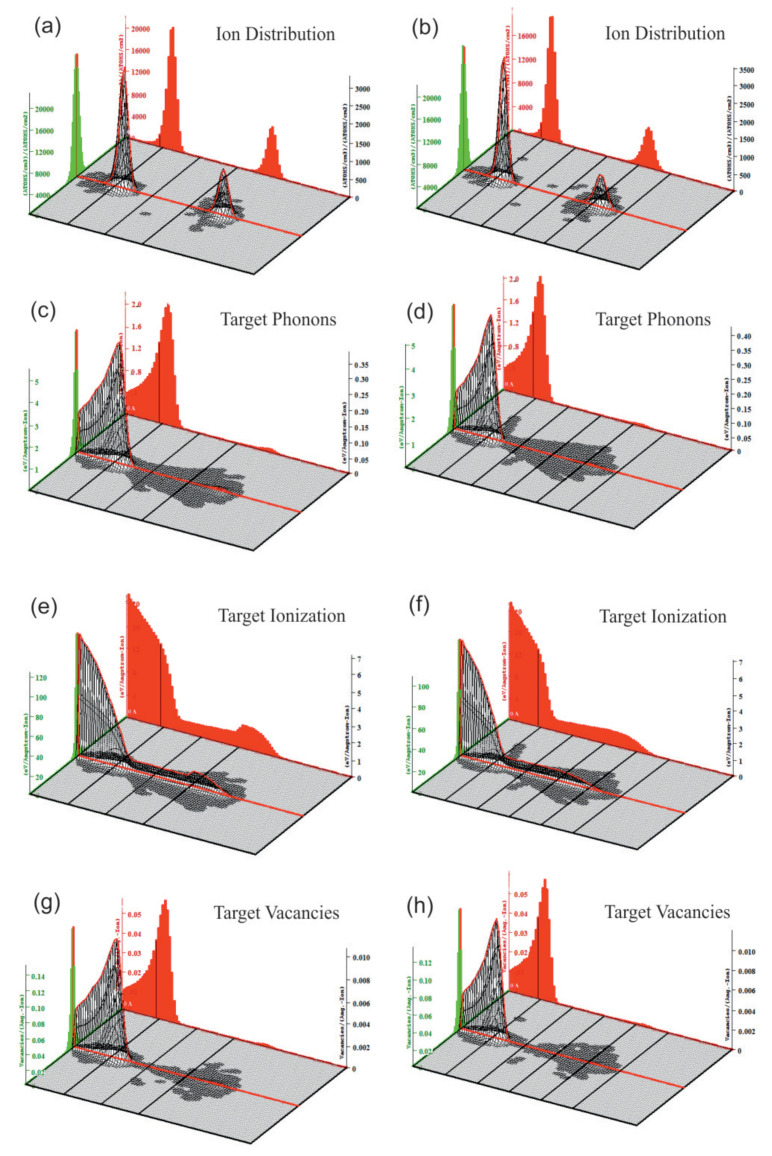
SRIM simulation of the interaction three- (left column) and five- (right column) layered structures deposited onto fused silica substrate with hydrogen and carbon ions beams. 3D spatial distribution at various depths across the surface of the target of the implanted hydrogen and carbon ions (**a**,**b**); energy losses due to phonon generation (**c**,**d**); energy losses due to ionization (**e**,**f**); energy losses due to phonons excitation; vacancies distribution (**g**,**h**). All dependencies are given per one incident ion. Energy losses distribution are given in eV/(Angstrom Ion) and vacancies distribution in vacancies/(Angstrom Ion). The lateral (green) and longitudinal (red) distributions of ions along corresponding axis are also shown.

**Figure 7 nanomaterials-11-00170-f007:**
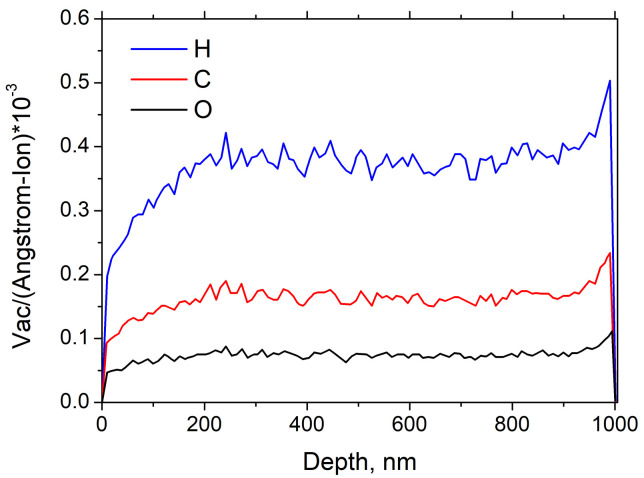
Vacancies distribution in 1 μm thick graphene/PMMA bi-layer for proton spectrum at geostationary orbit (GEO) (SRIM numerical simulation). In the PMMA spacer, the total numbers of H, C and O vacancies per incident ion are 1.84, 0.83 and 0.38, respectively. The total number of vacancies per one incident proton in graphene layer is about $2\times 10^{-3}$).

**Table 1 nanomaterials-11-00170-t001:** The results of the Stopping and Range of Ions in Matter package numerical simulation (SRIM-2013, http://www.srim.org/) of the number of vacancies produced in graphene/PMMA sandwich structures under irradiation with the ion beam consisting of carbon (70%) and hydrogen (30%) ions. Each bi-layer consists of 1 μm thick PMMA spacer and a graphene sheet. Table shows the number of vacancies per incident ion generated in the PMMA spacers, graphene sheets and SiO${}_{2}$ substrate for structures consisting of 1, 3 and 5 bi-layers.

Vacancy Type	Bi-Layer 1		Bi-Layer 2		Bi-Layer 3		Bi-Layer 4		Bi-Layer 5		SiO${}_{2}$ Substrate
	PMMA	Gr	PMMA	Gr	PMMA	Gr	PMMA	Gr	PMMA	Gr	
H	116.0										
C	56.5	10${}^{-4}$									
O	24.6										142.0
Si											139.0
**Total vac/ion**	**197.1**										**281.0**
H	109.3		120		0.18						
C	53.2	10${}^{-4}$	57	0	0.10	0					
O	23.4		24.5		0.04						2.5
Si											2.0
**Total vac/ion**	**185.9**		**201**		**0.32**						**4.5**
H	114.0		120.0		0.16		0.7		2.0		0
C	55.6	10${}^{-4}$	57.0	0	0.09	0	0.4	0	0.7	0	0
O	24.4		24.5		0.04		0.2		0.3		0
Si											0
**Total vac/ion**	**194.0**		**202**		**0.29**		**1.3**		**3.0**		**0**

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
