# Peer review of "Outstanding Radiation Tolerance of Supported Graphene: Towards 2D Sensors for the Space Millimeter Radioastronomy"

_nanomaterials, 2021, doi:10.3390/nano11010170_

Round 1

Reviewer 1 Report

This manuscript explores an interesting topic of the radiation tolerance of graphene. However, in my opinion, the present work has not provided convincing evidence to demonstrate the radiation tolerance for graphene.

(1) In experimental parts, investigated are only the radiation effects on the Raman spectra and transmittance of graphene/PMMA stacks. For Raman spectra, the authors only compare very qualitatively between the D peaks of the sample before and after radiation. However, Raman spectra itself can hardly provide full information about the defects (please see e.g., Nano Lett. 2011, 11, 8, 3190–3196). Even though Raman spectra are used for defect analysis, typically it is the intensity ratio of D peak to G peak, i.e., I_D/I_G, that is used to measure the defect density of graphene. As for the transmittance, it is found after radiation, the transmittance of the graphene/PMMA structure could increase. I am not sure how this performance can be related to the radiation tolerance of graphene. I thereby suggest the authors provide more experimental evidence. As they mentioned the graphene/PMMA stacks are used for bolometer, it could be more convincing to characterize and compare the performance of the bolometers before and after radiation.

(2) The simulation part shows the vacancy distribution along the depth of the graphene/PMMA stacks. However, it is still not easy to see how the graphene layers resist radiation. For example, in Fig. 7, the total vacancy per ion in 1-um-thick PMMA is 1.84 + 0.83 + 0.38 = 3.05, while that in graphene (0.34 nm thick?) is 2 X 10^-3. If one scales the thickness of PMMA down to 0.34 nm, the vacancy per ion is only 1.04 X 10^-3. Does that mean PMMA is more tolerant to radiation than graphene?

(3) It is better to perform “control simulations” where only PMMA layers are stacked without graphene to clearly indicate the function/effects of graphene.

(4) Finally, I see some inconsistency between the Introduction section and other section. In Introduction section, it looks the authors have accepted the good radiation resistance of graphene (based on the literature) but will investigate graphene/PMMA stacks in this paper. So it looks this work should focus on the radiation resistance of PMMA, but actually the title and other sections still focus on graphene.

Reviewer 2 Report

The manuscript by Paddubskaya et al. reports on investigations of the effect of radiation on the performance of graphene sensors for millimeterwave radioastronomy. The manuscript is a continuation of previous research by this group [13-15]. Here the devices are bombarded by proton and C ions. Raman scattering is used to verify the irradiation damage. At the end it turns out that a sort of ablation at the front layer is most important. It would be of high interest, to investigate the effect of electron irradiation, too ; in particular since this is most crucial in GEO.
The conclusion of the authors is: proton and C irradiation causes only little damage, and thus electrons will do even less. While this might be fine for an upper limit, it is not enough for a quantitative estimate.
Again, why not measure electron irradiation damage directly?

Minor remarks:
The figures are not explained sufficiently by the captions, e.g. the blooms in Fig. 5 are not labelled, the colors in Fig. 6 are not identified, etc..
There are numerous formal and orthographic mistakes, poor language etc. It is inevitable to carefully check the manuscript and ask professionals or native speakers for support.
